# Drug mechanism-of-action discovery through the integration of pharmacological and CRISPR screens

Emanuel Gonçalves[1], Aldo Segura-Cabrera[2], Clare Pacini[1], Gabriele Picco[1], Fiona M Behan[1], Patricia Jaaks[1], Elizabeth A Coker[1], Donny van der Meer[1], Andrew Barthorpe[1], Howard Lightfoot[1], Tatiana Mironenko[1], Alexandra Beck[1], Laura Richardson[1], Wanjuan Yang[1], Ermira Lleshi[1], James Hall[1], Charlotte Tolley[1] (iD), Caitlin Hall[1], Iman Mali[1], Frances Thomas[1], James Morris[1], Andrew R Leach[2], James T Lynch[3], Ben Sidders[3], Claire Crafter[3], Francesco Iorio[1,4] (iD), Stephen Fawell[5] & Mathew J Garnett[1,*] (iD)

## Abstract

Low success rates during drug development are due, in part, to the difficulty of defining drug mechanism-of-action and molecular markers of therapeutic activity. Here, we integrated 199,219 drug sensitivity measurements for 397 unique anti-cancer drugs with genome-wide CRISPR loss-of-function screens in 484 cell lines to systematically investigate cellular drug mechanism-of-action. We observed an enrichment for positive associations between the profile of drug sensitivity and knockout of a drug's nominal target, and by leveraging protein–protein networks, we identified pathways underpinning drug sensitivity. This revealed an unappreciated positive association between mitochondrial E3 ubiquitin–protein ligase *MARCH5* dependency and sensitivity to MCL1 inhibitors in breast cancer cell lines. We also estimated drug on-target and off-target activity, informing on specificity, potency and toxicity. Linking drug and gene dependency together with genomic data sets uncovered contexts in which molecular networks when perturbed mediate cancer cell loss-of-fitness and thereby provide independent and orthogonal evidence of biomarkers for drug development. This study illustrates how integrating cell line drug sensitivity with CRISPR loss-of-function screens can elucidate mechanism-of-action to advance drug development.

**Keywords** CRISPR-Cas9; drug mechanism-of-action; protein networks
**Subject Category** Pharmacology & Drug Discovery
**Mol Syst Biol. (2020) 16: e9405**

## Introduction

Understanding drug mechanism-of-action and evaluating in cellular activity is challenging (Santos *et al*, 2017), and widespread target promiscuity contributes to low success rates during drug development (Klaeger *et al*, 2017). For target-based drug development, a detailed understanding of drug mechanism-of-action provides information about specificity and undesirable off-target activity which could lead to toxicity and a reduced therapeutic window (Lin *et al*, 2019). Furthermore, molecular biomarkers can be used to monitor drug activity for patient stratification during clinical development.

The cellular activity of a drug is influenced by multiple factors including the affinity and selectivity of the compound for its target (s) and the penetrance of target engagement on cellular phenotypes. An array of biochemical, biophysical, computational and cellular assays are used to investigate drug mechanism-of-action (Schenone *et al*, 2013). For example, protein kinase inhibitors are profiled *in vitro* for their specificity and potency against panels of purified recombinant protein kinases. While informative, this approach fails to recapitulate the native context of the full-length protein in cells which could influence true drug activity, nor does it identify potential non-kinase off-target effects. Cellular-based approaches to investigate mechanism-of-action include transcriptional profiling following drug treatment of cells, chemical proteomics approaches such as kinobeads (Bantscheff *et al*, 2007; Médard *et al*, 2015) and cellular thermal shift assay (Savitski *et al*, 2014) to measure drug–protein interactions, and multiplexed imaging or flow-cytometry to measure multiple cellular parameters upon drug treatment (Li *et al*, 2017; Subramanian *et al*, 2017; Reinecke *et al*, 2019). Despite the utility of these different approaches, gaining a full picture of drug mechanism-of-action, particularly in cells, remains a challenge and new approaches would be beneficial.

1   Wellcome Sanger Institute, Hinxton, UK
2   European Molecular Biology Laboratory, European Bioinformatics Institute, Hinxton, UK
3   Research and Early Development, Oncology R&D, AstraZeneca, Cambridge, UK
4   Human Technopole, Milano, Italy
5   Research and Early Development, Oncology R&D, AstraZeneca, Waltham, MA, USA
   *Corresponding author. Tel: +44 (0)1223 494878; E-mail: mg12@sanger.ac.uk

Pharmacological screens (Barretina *et al*, 2012; Garnett *et al*, 2012; Iorio *et al*, 2016; Subramanian *et al*, 2017; Lee *et al*, 2018) have been used to profile the activity of hundreds of compounds in highly annotated collections of cancer cell lines with the aim of identifying molecular markers of drug sensitivity to guide clinical development (Cook *et al*, 2014; Nelson *et al*, 2015). More recently, CRISPR-based gene-editing has enabled the evaluation of highly specific and penetrant gene knockout effects on cell fitness genome-wide in hundreds of cancer cell lines (Jinek *et al*, 2012; Shalem *et al*, 2014; Hart *et al*, 2015; Meyers *et al*, 2017; Behan *et al*, 2019). This has provided rich functional resources to explore cancer vulnerabilities and to identify candidate drug targets (Marcotte *et al*, 2016; Meyers *et al*, 2017; Tsherniak *et al*, 2017; Behan *et al*, 2019). Parallel integration of gene loss-of-function screens with drug response can be used to investigate drug mechanism-of-action (Deans *et al*, 2016; Subramanian *et al*, 2017; Jost & Weissman, 2018; Wang *et al*, 2018; Zimmermann *et al*, 2018; Hustedt *et al*, 2019a,b).

Here, we integrate recent genome-wide CRISPR-Cas9 loss-of-function screens with pharmacological data for 397 unique anti-cancer compounds in 484 cancer cell lines. This provided an unsupervised and comprehensive characterisation of drug mechanism-of-action using CRISPR-Cas9 screens and recapitulated nominal drug targets, gave insights into drug potency and selectivity, and defined cellular networks underpinning drug sensitivity. Illustrating the utility of our approach, we identified a link between mitochondrial ubiquitin ligase *MARCH5* in the response to MCL1 inhibitors, especially in breast cancer cell lines. Furthermore, we defined robust pharmacogenomic associations, represented by genetic biomarkers independently supported by drug response and gene fitness measurements. These identify genetic contexts associated with drug-pathway dependency and provide a more refined set of biomarkers. Collectively, we present an approach to leverage pharmacological and CRISPR screening data to inform on drug in cellular mechanism-of-action and thus guide drug development.

## Results

### Cancer cell line drug sensitivity and gene fitness effects

We analysed data sets from a highly annotated collection of 484 histologically diverse human cancer cell lines (Dataset EV1). Cell line information is available through the Cell Model Passports web portal (http://cellmodelpassports.sanger.ac.uk/) (van der Meer *et al*, 2019). These have been extensively genetically characterised and utilised for both large-scale drug sensitivity testing and CRISPR-Cas9 whole-genome loss-of-function screens (Garnett *et al*, 2012; Iorio *et al*, 2016; Meyers *et al*, 2017; van der Meer *et al*, 2019; Picco *et al*, 2019) as part of the Cancer Dependency Map initiative (https://depmap.sanger.ac.uk/). We expanded on published single agent drug sensitivity data (Garnett *et al*, 2012; Iorio *et al*, 2016; Lynch *et al*, 2016; Picco *et al*, 2019) to consider 199,219 $IC_{50}$ values for 397 unique cancer drugs (480 drugs including duplicates, Dataset EV2). These encompassed FDA-approved cancer drugs, drugs in clinical development and investigational compounds, with multiple modes of action, including 24 chemotherapeutic agents and 367 small molecule inhibitors. Drugs considered in this study had a

response in at least three cell lines ($IC_{50}$ lower than half of the maximum screened concentration) and 86% of all possible drug/cell line $IC_{50}$ measurements have been experimentally evaluated (Fig EV1A, Dataset EV3). Two experimental protocols were used to generate drug sensitivity measurements, named GDSC1 (Iorio *et al*, 2016) and GDSC2 (Picco *et al*, 2019) (Fig EV1B). A principal component analysis (PCA) of $IC_{50}$ values identified a screen specific batch effect associated with principal component (PC) 2 which explained 2.8% of the total variance (Fig EV1C). For this reason, despite the fact that compounds screened with both technologies showed good agreement ($n = 66$, mean Pearson's $R = 0.50$), we analysed the measurements of the screens separately. Analysis of the drug response variation across cell lines revealed that PC 1 (28.7% variance captured) was significantly and negatively correlated with cell line growth rate (Pearson's $R = -0.51$, $P$-value = 1.2e-28), particularly for chemotherapy agents and cell cycle inhibitors (Figs EV1D and E).

Cell fitness effects for 16,643 gene knockouts have been measured using genome-wide CRISPR-Cas9 screens at the Sanger and Broad Institutes (Meyers *et al*, 2017; Behan *et al*, 2019; DepMap Broad, 2019) (Dataset EV4). The first PC across the cell lines (6.8% variance explained) separated the two institutes of origin (Fig EV2A), consistent with a comparative analysis performed on an overlapping set of cell lines (Dempster *et al*, 2019). Growth rate was less significantly associated with CRISPR knockout response (Figs EV2B and C).

### Gene knockout fitness effects correspond with drug targets

We began by investigating the extent to which drug sensitivity corresponded to CRISPR knockout of drug targets. In an unsupervised way, we systematically searched for all possible associations between the profile of drug sensitivity ($n = 480$) and gene fitness effects ($n = 16,643$) across the 484 cell lines (Fig 1A). We expect this to capture a variety of relationships ranging from direct drug–target interactions to more complex associations arising from interactions with regulators of the drug target(s). We tested a total of 7,988,640 single-feature gene–drug associations using linear mixed regression models. Potential confounding effects such as growth rate, culture conditions, data source and sample structure were considered in the models. We identified 865 significant associations (FDR-adjusted $P$-value < 10%, Dataset EV5) between drug response and gene fitness profiles (Fig 1B), termed hereafter as significant drug–gene pairs. For this analysis, we were able to manually curate the nominal therapeutic target(s) for 94.7% ($n = 376$) of the anti-cancer drugs screened (Fig EV3A and Dataset EV1).

For 26% ($n = 94$) of the 358 drugs with target annotation and for which the target was knocked-out with the CRISPR-Cas9 library, we identified significant drug–gene pairs with their putative targets (Fig 1C). For example, there were strong associations between MCL1 and BCL2 inhibitor sensitivity and their respective gene knockouts (Fig 1D). Notably, drug–gene associations with the drug target had a skewed distribution towards positive effect sizes (Mann–Whitney *U*-test $P$-value < 1.36e-105, Fig EV3B) and were amongst the strongest associations (Fig 1B). To investigate this further, we utilised independently acquired kinobead drug–protein affinity measurements for an overlapping set of 64 protein kinase inhibitors which were profiled for their specificity against 202

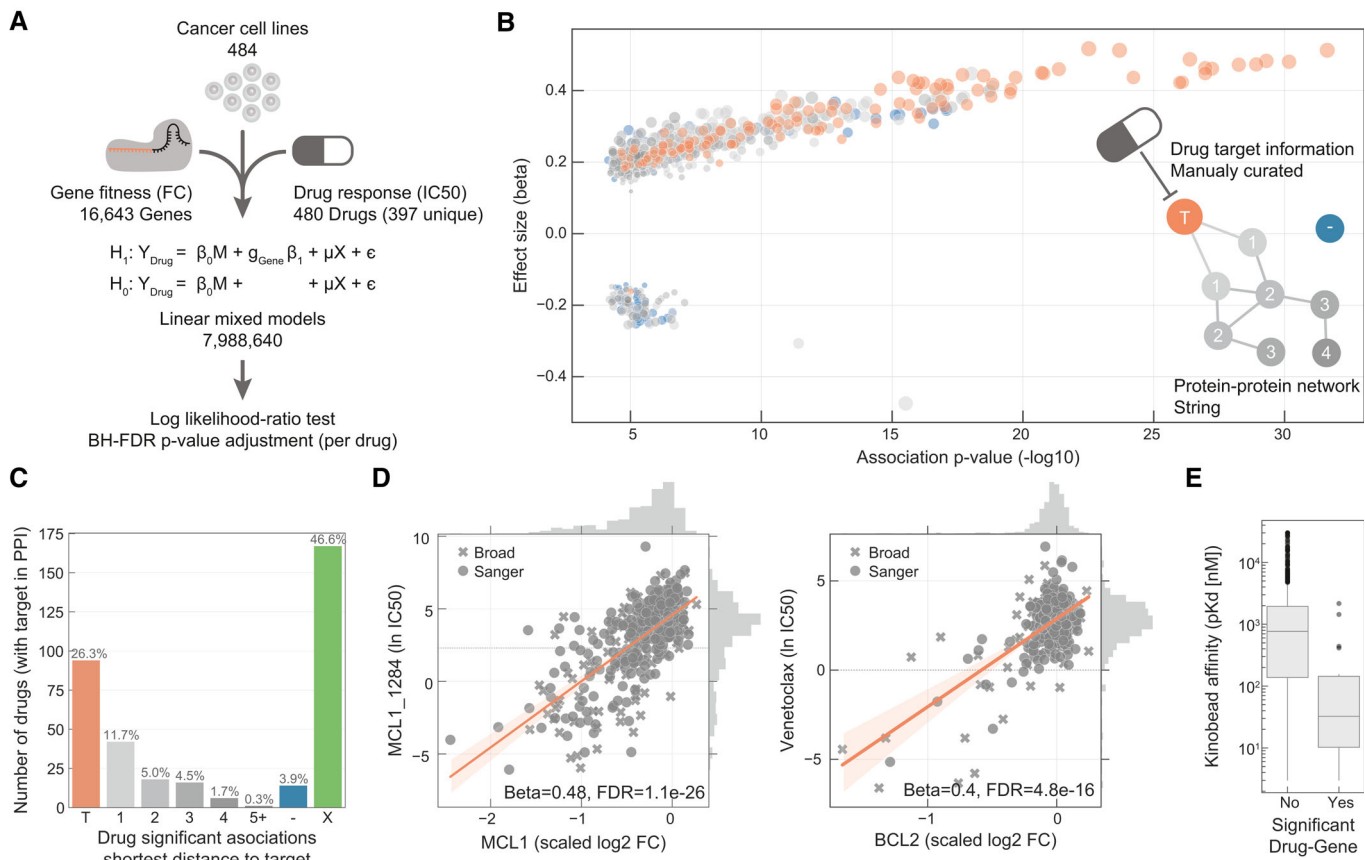

**Figure 1. Integration of drug and CRISPR gene dependencies in cancer cell lines.**

A   Linear models were used to integrate drug sensitivity (IC$_{50}$ values) and gene fitness measurements.

B   Volcano plot showing the effect sizes and the *P*-value for statistically significant associations, Benjamini–Hochberg false discovery rate (FDR)-adjusted likelihood-ratio test *P*-value < 10%. Drug–gene associated pairs are coloured according to their shortest distance in a protein–protein interaction network of the gene to any of the nominal target of the drug.

C   Percentage of the 358 drugs with significant associations and their shortest distance in the PPI network to the drug nominal targets. T represents drugs that have a significant association with at least one of their canonical targets, "—" represents no link was found, and X are those which have no significant association.

D   Examples of the top drug response correlations with target gene fitness. Each point represents an individual cell line. MCL1_1284 and venetoclax are MCL1 and BCL2 selective inhibitors, respectively. Gene fitness log$_2$ fold changes (FC) are scaled by using previously defined sets of essential (median scaled log$_2$ FC = −1) and non-essential (median scaled log$_2$ FC = 0) genes. Drug response IC$_{50}$ measurements are represented using the natural log (ln IC$_{50}$).

E   Kinobead affinity is significantly higher (lower *pK$_d$*) for compounds with a significant association with their target (*n* = 20, Mann–Whitney *P*-value = 3.1e-07). Box-and-whisker plots show 1.5× interquartile ranges and 5–95$^{th}$ percentiles, centres indicate medians.

kinases (Klaeger *et al*, 2017). Drugs with significant associations with knockout of their target also had stronger affinity to their target in the kinobead assay, providing independent evidence that the strongest drug–gene associations are enriched for targets of the drugs (Fig 1E). Overall, we identified the nominal target of approximately one-quarter of the drugs tested using orthogonal CRISPR gene fitness screens, and drug targets were amongst the most significant gene–drug associations.

**Cellular networks underpinning drug response**

For the remaining drugs (*n* = 264) which were not significantly associated with the CRISPR loss-of-function measurements of their nominal targets (Fig 1C), we reasoned that superimposing the significant drug–gene pairs onto a protein interaction network may shed further insights into the relationship of the associations. We used a protein–protein interaction (PPI) network assembled from the STRING database (Szklarczyk *et al*, 2017) (10,587 nodes and 205,251 interactions), and for the significant drug–gene pairs calculated the minimal distance between the drug nominal target(s) and the associated gene. Out of the 264 drugs, 76 drugs had a significant association with their target's first neighbour or a protein closely related in the network (1, 2 or 3 PPI distance from drug targets; Fig 1B and C). Thus, despite these drugs not showing significant associations with their nominal targets, CRISPR associations revealed potential mechanisms-of-actions which are functionally related to their targets. Taken together, for the 358 drugs with target annotation and covered by the CRISPR-Cas9 screens, 47.5% (*n* = 170) had an association with either the target (26.3%) or a functionally related protein (21.2%; Fig 1C).

The strongest drug–gene pair associations were between a drug and its canonical target(s) rather than components of the PPI

network, and significance decreased (along with the number of associations) as the interaction distance increased (Fig 2A). To exclude the possibility that this observation is biased by the topology of the network, we calculated the length of all the shortest paths between the drug target(s) and their associated genes and confirmed the enrichment of first and second neighbours in significant drug–gene associations (Fig EV3C). In comparison, cell line gene expression identified considerably fewer associations with the PPI neighbours of the drug target (Fig 2B; Dataset EV6). In particular, the number of drugs significantly associated with their targets substantially decreased ($n = 17$), and significant associations were predominantly found with proteins further away in the PPI network, close to the average length of all paths ($l_G = 3.9$). As an example, MIEN1 gene expression is positively correlated with multiple EGFR and ERBB2 inhibitors, which can be explained, not by a functional relationship, but more likely by genomic co-localisation with *ERBB2* on chromosome 17. Hence, CRISPR-Cas9 screens are powered to discriminate genes which are likely to be functionally linked to drug response.

To investigate putative regulatory networks for drugs, we weighted the PPI network edges with the correlation between the fitness profiles of the two connected nodes and integrated the resulting weighted network with drug response associations. *EGFR* inhibitors are the most abundant drug class in our set, and we observed that multiple inhibitors (e.g. cetuximab) showed significant associations with *EGFR* and known pathway members, for example *SHC1* and *GRB2* (Scaltriti & Baselga, 2006; Zheng *et al*, 2013) (Fig 2C). Additionally, the weighted network shows pathway members that have strongly correlated fitness profiles, which are likely functionally related (Wang *et al*, 2017a,b; Boyle *et al*, 2018; Pan *et al*, 2018; Rauscher *et al*, 2018; Kim *et al*, 2019). For EGFR inhibitors, these included receptor tyrosine kinase *MET* and the protein phosphatase *PTPN11* (Wang *et al*, 2017a; Pan *et al*, 2018) (Fig 2D). Drug–target tailored networks can be used to understand drug mechanism-of-action and have the potential to identify resistance mechanisms and thus can be used to identify new alternative drug targets. Collectively, our network analysis demonstrates that CRISPR screens can provide functional insights into drug in cellular activity, extending beyond the direct drug target, into the associated functional network.

Despite our finding that we can illuminate drug functional networks, 46.6% ($n = 167$) of the tested drugs had no significant drug–gene associations. A number of possible technical and biological factors may underpin this observation. In support of a link between drug selectivity and significant gene–drug associations (Fig 1B and E), drugs with no significant association with their target had in general a higher number of putative targets based on ChEMBL bioactivity profiles (two-sided Welch's t-test

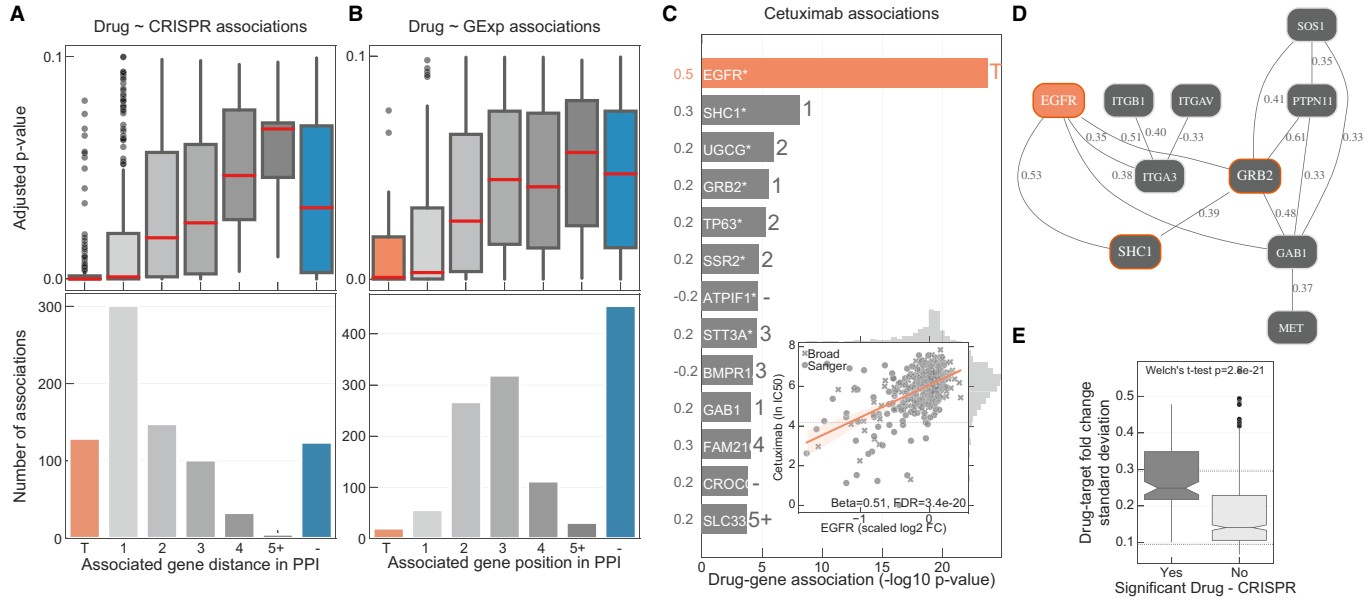

**Figure 2. Drug response protein–protein networks.**

A    Distribution of the FDR-adjusted *P*-values (top) and count (bottom) of the significant (FDR-adjusted likelihood-ratio test *P*-value < 10%) drug–gene (CRISPR) associations according to their distance between the gene and corresponding drug targets in the protein–protein interaction network. T represents drugs that have a significant association with at least one of their canonical targets and "−" represents no link was found. Box-and-whisker plots show 1.5× interquartile ranges and 5–95th percentiles, centres indicate medians.

B    Similar to (A), but instead gene expression (GExp) was tested to identify associations with drug response. T represents drugs that have a significant association with at least one of their canonical targets and "−" represents no link was found. Box-and-whisker plots show 1.5× interquartile ranges and 5–95th percentiles, centres indicate medians.

C, D  (C), Representative example, i.e. cetuximab—EGFR inhibitor, of the associations and (D), networks that can be obtained from the integrative analysis. Edges in the network are weighted with the Pearson correlation coefficient obtained between the fitness profiles of interacting nodes. For representation purposes only edges with the highest correlation coefficient were represented, $R^2 > 0.3$. Nodes with orange borders represent significant associations with drug response, cetuximab.

E    Drug–target associations grouped by statistical significance (FDR-adjusted likelihood-ratio test *P*-value < 10%) and plotted against the standard deviation of the drug–target CRISPR fold changes (significant "Yes" $n = 129$, significant "No" $n = 684$). Upper and lower dashed lines represent the standard deviations of essential and non-essential genes, respectively. Box-and-whisker plots show 1.5× interquartile ranges and 5–95th percentiles, centres indicate medians.

*P*-value = 0.003; Fig EV3D), and drugs with no significant associations were approximately three times less likely to be associated with a genomic biomarker linked to sensitivity (Fig EV3E). Alternative explanations include lower variability in CRISPR fold change measurements for the target of these drugs (Fig 2E). For example, where genetic knockout of an essential gene induces a strong loss-of-fitness effect uniformly across all cell lines, whereas a drug has more variable effects likely due to incomplete target inhibition (Fig EV3F). The lack of variability was much less pronounced in the drug sensitivity measurements since we only considered drugs which showed a minimal level of activity, i.e. $IC_{50}$ lower than half of the maximum screened concentration (Fig EV3G). Inhibition of a protein is also intrinsically different than a knockout, as observed for PARP inhibitors whose activity is mediated through formation of cytotoxic PARP-DNA complexes, whereas PARP knockout has little or no effect on cell fitness (Gill *et al*, 2015; Murai & Pommier, 2015; Antolin *et al*, 2020) (Fig EV3H). Similarly, redundancy of gene paralogs when using single-gene CRISPR knockout may confound comparisons with drugs that target multiple paralogs (preprint: Dede *et al*, 2020). Thus, although the reasons may vary for each drug, the absence of an association between drug sensitivity and CRISPR loss-of-function effects could warrant further investigation into drug mechanism-of-action to understand possible underlying factors, such as low potency, alternative molecular mechanisms, or polypharmacology.

**Cancer drug mechanism-of-action**

Next, we set out to investigate in detail some of the strongest drug sensitivity and CRISPR gene fitness associations (Dataset EV5). Strikingly, 46 of the top 50 strongly associated drugs had significant associations with their nominal target and with known functionally related genes. Some of the strongest associations were between MCL1 inhibitors and their target (Fig 3—MCL1 and BLC2 inhibitors panel), including AZD5991 which is currently in clinical trials for treatment of haematologic cancers (Hird *et al*, 2017). Additionally, for several insulin-like growth factor 1 receptor (IGFR1) inhibitors the association with the target was recapitulated. Moreover, significant associations with proprotein convertase *furin* were observed, supporting the genetic association that IGFR1 is a furin substrate. Moreover, increased levels of furin are associated with increased levels of processed IGFR1 and worse prognosis in several cancers (Thomas, 2002).

The clinical development of kinase inhibitors is hampered by poor selectivity because of the conserved structural features of the commonly targeted kinase domain, which could lead to undesirable off-target activity (Klaeger *et al*, 2017). Furthermore, some kinases have multiple isoforms with non-redundant roles. For example, isoform-selective PI3K inhibitors have been developed in part to reduce toxicity and increase the therapeutic window (Thorpe *et al*, 2015). Interestingly, several PI3K inhibitors had strong associations with only one gene encoding a single isoform (Fig 3—PIK3C inhibitors panel). This together with the increased kinobead binding affinity of significant associations (Fig 1E) suggests these are isoform-selective compounds. For example, alpelisib was associated with *PIK3CA*, consistent with its development as an alpha-isoform-selective compound (Thorpe *et al*, 2015), whereas AZD8186 was only associated with *PIK3CB* confirming its beta-selectivity. Conversely, two *pan*-PI3K inhibitors (buparlisib and omipalisib) displayed no

significant association with any *PI3K* isoform (Dataset EV5), consistent with its lack of isoform specificity and potential polypharmacology. Interestingly, MTOR and pan-PI3K inhibitor, dactolisib, had significant associations with *RPTOR* and *MTOR* but none with *PI3K* isoforms (Dataset EV5), consistent with recently reported greater specificity for inhibition of the MTOR complex (Reinecke *et al*, 2019). Similarly, we observed that selective EGFR inhibitors cetuximab, erlotinib and gefitinib (Fig 3—EGFR inhibitors panel) were associated with *EGFR* but not *ERBB2*, whereas sapatinib, afatinib and poziotinib (Fig 3—ERBB2; EGFR inhibitors panel) were all associated with both *EGFR* and *ERBB2*.

Our analysis can also provide insights into possible off-target activity of drugs. Unsupervised clustering of the drug–gene associations effect sizes (betas) revealed classes of inhibitors with similar targets and mechanism-of-action (Fig EV3I). Of note, BTK inhibitor, ibrutinib, clustered with EGFR inhibitors and displayed significant associations with *EGFR* and *ERBB2* gene fitness (Fig 3—EGFR inhibitors panel). This is consistent with recent findings that ibrutinib covalently binds to and inhibits EGFR (Lee *et al*, 2018) and is also supported by kinobead measurements (Klaeger *et al*, 2017). Additionally, 24 compounds have significant associations with genes identified as essential core fitness (Behan *et al*, 2019) across multiple cancer types, indicating an increased risk of cellular toxicity. Out of these, two compounds, PD0166285 and CCT244747, have significant associations with their nominal target (*PKMYT1* and *CHEK1/WEE1*) and the remaining compounds (*n* = 22) are correlated with core fitness proteins closely connected in the PPI network.

**A functional link between MARCH5 and MCL1 inhibitor sensitivity**

Seven out of nine unique inhibitors of the anti-apoptotic *BCL2* family member myeloid cell leukaemia 1 (MCL1) were strongly and nearly exclusively associated with their putative target, suggesting these are potent and specific compounds in cells (Fig 4A). *MCL1* is frequently amplified in human cancers (Beroukhim *et al*, 2010) and associated with chemotherapeutic resistance and relapse (Wuillème-Toumi *et al*, 2005; Wei *et al*, 2006). MCL1 is a negative regulator of the mitochondrial apoptotic pathway, regulating *BAX/BAK1* which co-localise with Drp1/Fis1 in the mitochondria outer membrane and control mitochondrial fragmentation and cytochrome c release, both of which are important for inducing apoptosis (Youle & Karbowski, 2005; Mojsa *et al*, 2014; Morciano *et al*, 2016). Interestingly, knockout of a key regulator of mitochondrial fission, mitochondrial E3 ubiquitin-protein ligase *MARCH5 (Karbowski et al, 2007)*, is significantly associated with MCL1 inhibitors sensitivity (Fig EV4A) and positively correlated with MCL1 gene fitness, suggesting a functional relationship (Fig 4B). Correlation between *MCL1* and *MARCH5* fitness profiles shows that cell lines dependent on *MARCH5* are also dependent on *MCL1*, while the inverse is not necessarily true with a subgroup of cell lines dependent on *MCL1* but not on *MARCH5* (Fig 4B). Cell lines independently dependent on both gene products have increased sensitivity to MCL1 inhibitors (Fig EV4B). This is particularly marked in breast carcinoma cancer cell lines, with *MCL1*- and *MARCH5*-dependent cells having similar sensitivity to haematologic cancer cell lines (acute myeloid leukaemia), where MCL1 inhibitors are in clinical development (Fig 4C).

We investigated the molecular mechanisms underlying the responses of MCL1 inhibitors. *MCL1* copy number and gene

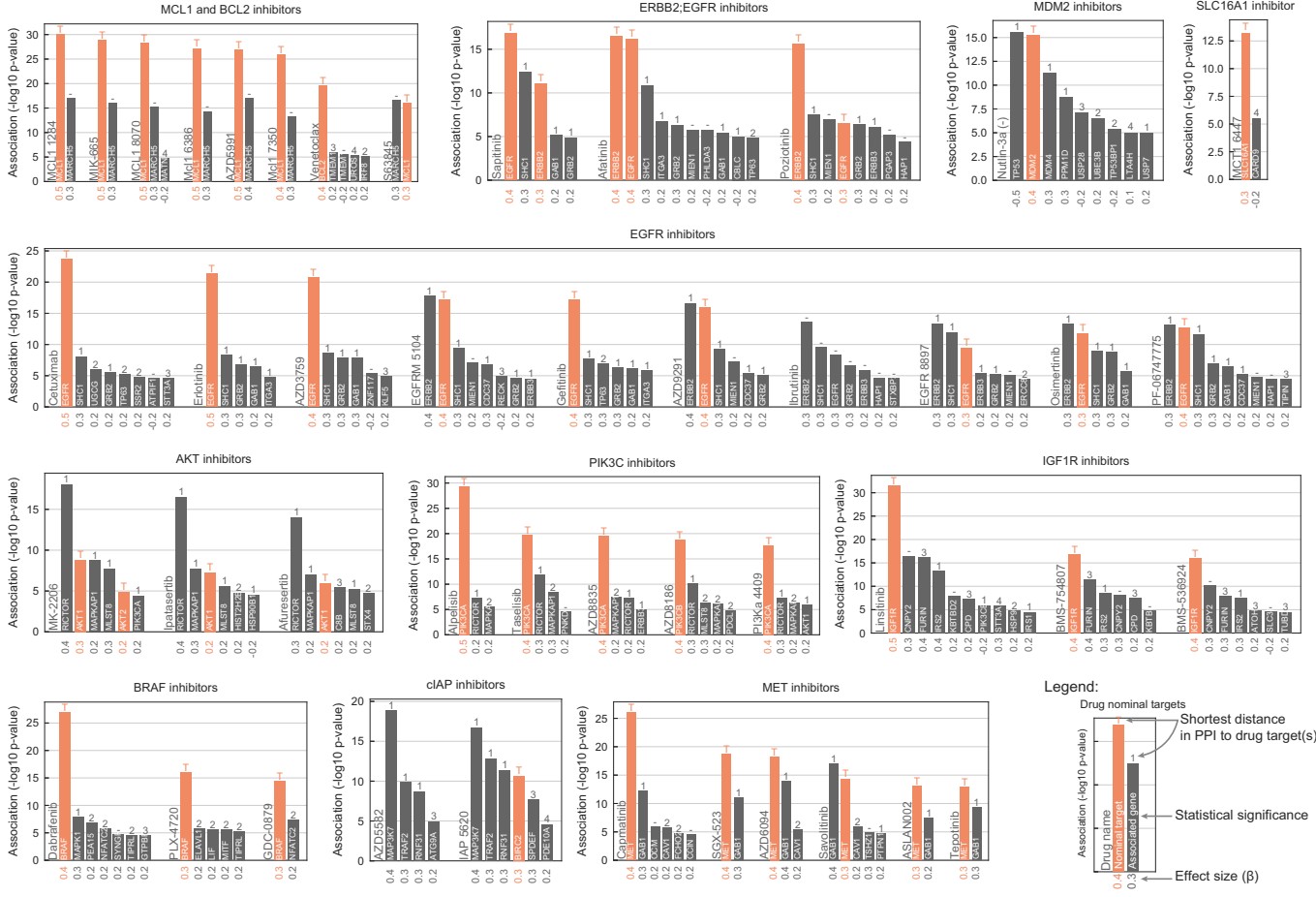

**Figure 3. Representative examples of top most significantly associated drug classes.**

Each bar plot group represents a unique drug where genes are ranked by statistical significance (likelihood-ratio test P-value) of their association. Effect sizes of the associations are reported under the bars along the x axis. Shortest distance (number of interactions) in a protein–protein interaction network between the gene and the drug nominal target(s) is represented on the top of the bars, where T and orange bar represent the target and "–" represents no link was found.

expression alone are not a good predictor of MCL1 inhibitors sensitivity (Figs 4D and EV4C). This is in contrast to BCL2 and BCL2L1 inhibitors, where their target gene expression is significantly correlated with drug sensitivity (Fig 4D). Next, we used multilinear regression models to predict sensitivity to each MCL1 inhibitor using gene fitness and/or gene expression of known regulators of *MCL1* (e.g. *BCL2*, *BCL2L1*, *BAX*) (Czabotar et al, 2014) and *MARCH5*. For two MCL1 inhibitors, MIM1 and UMI-77, the trained models performed poorly, likely due to lack of in cellular activity of these compounds as suggested by their drug response poor correlation with MCL1 gene essentiality profiles (Figs 4A and EV4A). For the remaining seven MCL1 inhibitors, drug response was well predicted (CRISPR + GEXP mean $R^2$ = 0.55). Models trained with only CRISPR displayed overall better predictions compared to models only trained with gene expression, and models trained with both data types out-performed all others (Fig 4E). As expected, MCL1 fitness effect was the most predictive feature, followed by decreased BCL2L1 expression and increased MARCH5 essentiality (Fig 4F). No genomic feature, mutation or copy number alterations correlated significantly with MCL1 inhibitors response, including *MCL1* amplifications (Fig EV4C), likely a consequence of the strong

post-transcriptional regulation and short half-life of the MCL1 protein.

Altogether, these results highlight a functional link between *MARCH5* and sensitivity to MCL1 inhibitors. This is consistent with recent reports of a synthetic-lethal interaction between *MARCH5* and the *MCL1* negative regulator *BCL2L1* (DeWeirdt et al, 2020), and MARCH5-dependent degradation of the MCL1/NOXA complex (Djajawi et al, 2020; Haschka et al, 2020). With further investigation, the link between MCL1 and MARCH5 could shed light on the mechanism-of-action of MCL1 inhibitors and the development of stratification approaches in solid tumours, such as breast carcinomas.

## Robust molecular markers of drug sensitivity networks

The identification of molecular biomarkers of drug sensitivity is fundamental to guide drug development. We hypothesised that molecular biomarkers independently linked with both drug response and gene fitness would be of particularly high value—termed robust pharmacogenomic biomarkers. To identify these, we used the set of significant drug–gene pairs ($n$ = 865) and we searched independently for significant associations between each measurement type

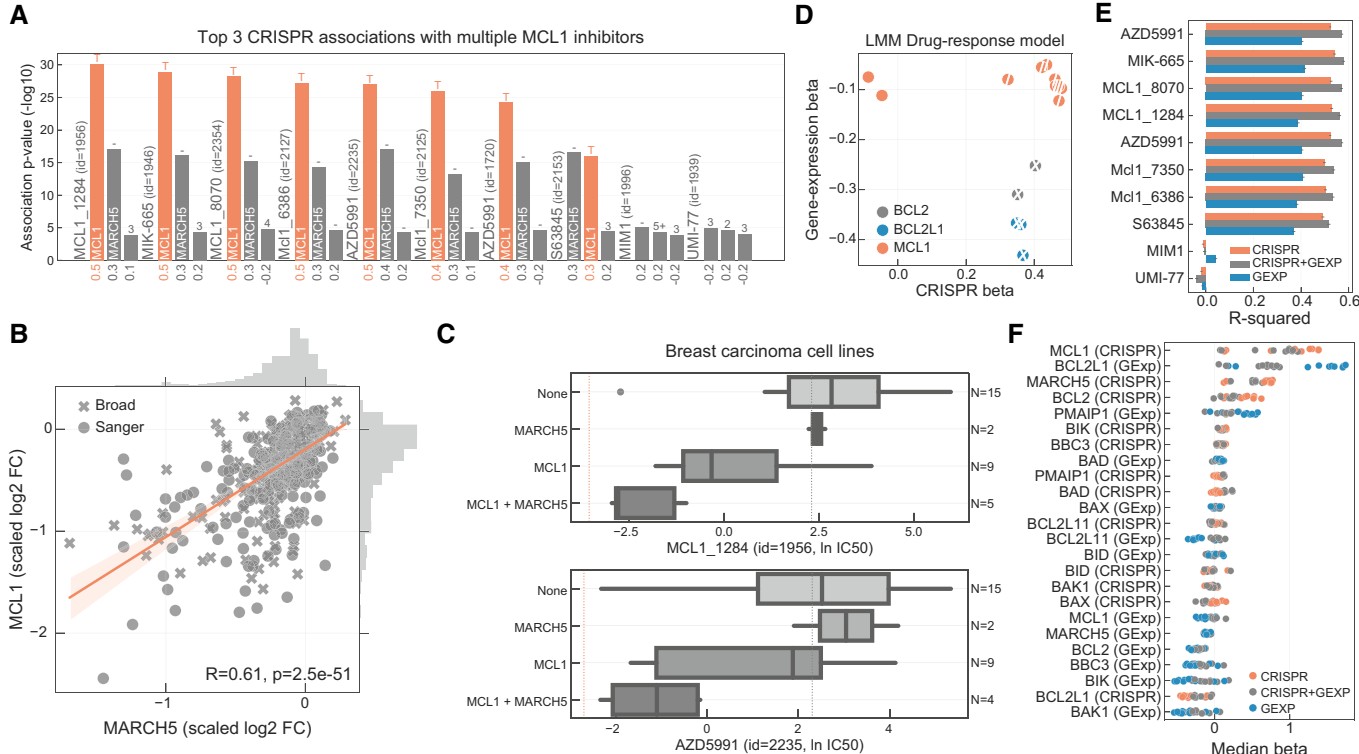

**Figure 4. MCL1 inhibitors associations.**

A   Top three CRISPR associations with all MCL1 inhibitors screened. Each bar represents the likelihood-ratio test *P*-value of each drug–gene (CRISPR) association and effect sizes reported under the bars. Shortest distance (number of interactions) in a protein–protein interaction network between the gene and the drug nominal target(s) is represented on the top of the bars, where T and orange bar represent the target and "−" represents no link was found.

B   Association between the gene fitness profiles of MCL1 and MARCH5.

C   Stratification of the MCL1 inhibitor sensitivity according to the essentiality profile of MCL1 and MARCH5, where MCL1 + MARCH5 represents a cell line that is independently dependent on both genes. Dashed orange line (left) represents the mean IC$_{50}$ in acute myeloid leukaemia cell lines. Grey dashed line (right) represents the maximum concentration used in the dosage response curve. Box-and-whisker plots show 1.5× interquartile ranges and 5–95$^{th}$ percentiles, centres indicate medians.

D   BCL2, BCL2L1 and MCL1 inhibitors and the respective association with their targets, on the *x* axis with CRISPR gene fitness and on the *y* axis with gene expression. The statistical significance (FDR-adjusted likelihood-ratio test *P*-value < 10%) of the association is represented with a backward slash for CRISPR and forward slash for GEXP.

E   Regularised multilinear regression to predict drug response of all MCL1 inhibitors using gene expression, fitness or both of known regulators of the BCL2 family and MARCH5. Predictive performance is estimated using $R^2$ metric represented in the *x* axis.

F   Effect size of each feature used in each MCL1 inhibitor model.

(drug response or gene fitness) and 519 genomic (mutations and copy number alterations) and 15,368 gene expression features (Figs 5A and EV5A) (Garnett *et al*, 2012; Iorio *et al*, 2016; Garcia-Alonso *et al*, 2018). This analysis recapitulated established genomic and expression biomarkers of either drug sensitivity or gene fitness effects in cancer cells (Figs EV5B and C). A total of 224 and 679 robust pharmacogenomic associations (FDR < 10%) were identified with genomic (Dataset EV7) and gene expression features (Dataset EV8), respectively. Overall, 30.6% (265 of 865) of drug–gene pairs have at least one robust molecular marker that correlated significantly with both drug response and gene fitness (Fig 5B). The number of robust pharmacogenomic biomarkers was smaller than the number of biomarkers associated with only one type of measurement, likely due to the stringent requirement for an independent association with both drug sensitivity and gene fitness effects.

From the subset of 129 drug–gene pair associations that were linked by the drug target, 50.4% (*n* = 65) had one or more robust

pharmacogenomic associations (Fig EV5D). Most of these were established dependencies of cancer cells, including Nutlin-3a sensitivity and MDM2 gene fitness independently associated with *TP53* mutation status; *BRAF* and *PIK3CA* mutation induced drug and CRISPR dependency; olaparib sensitivity mediated by the presence of *EWSR1-FLI1* fusion, also recapitulated by *FLI1* essentiality profile; MCL1 inhibitor and gene fitness associations with BCL2L1 expression, and Nutlin-3a sensitivity and MDM2 gene fitness associated with BAX expression (Figs 5C and EV5E and F). Similarly, of the 413 significant gene–drug pairs closely related within the PPI network (≤ 3 interactions from the drug target), we identified robust pharmacogenomic associations for 29.5% (*n* = 122; Fig EV5D), enabling the discovery of cellular contexts where drug response networks are important. For example, we identified increased tumour necrosis factor (TNF) expression as a robust pharmacogenomic marker for drugs targeting the downstream cellular inhibitor of apoptosis (cIAP) proteins *BIRC2* and *BIRC3* (e.g. IAP_5620), and

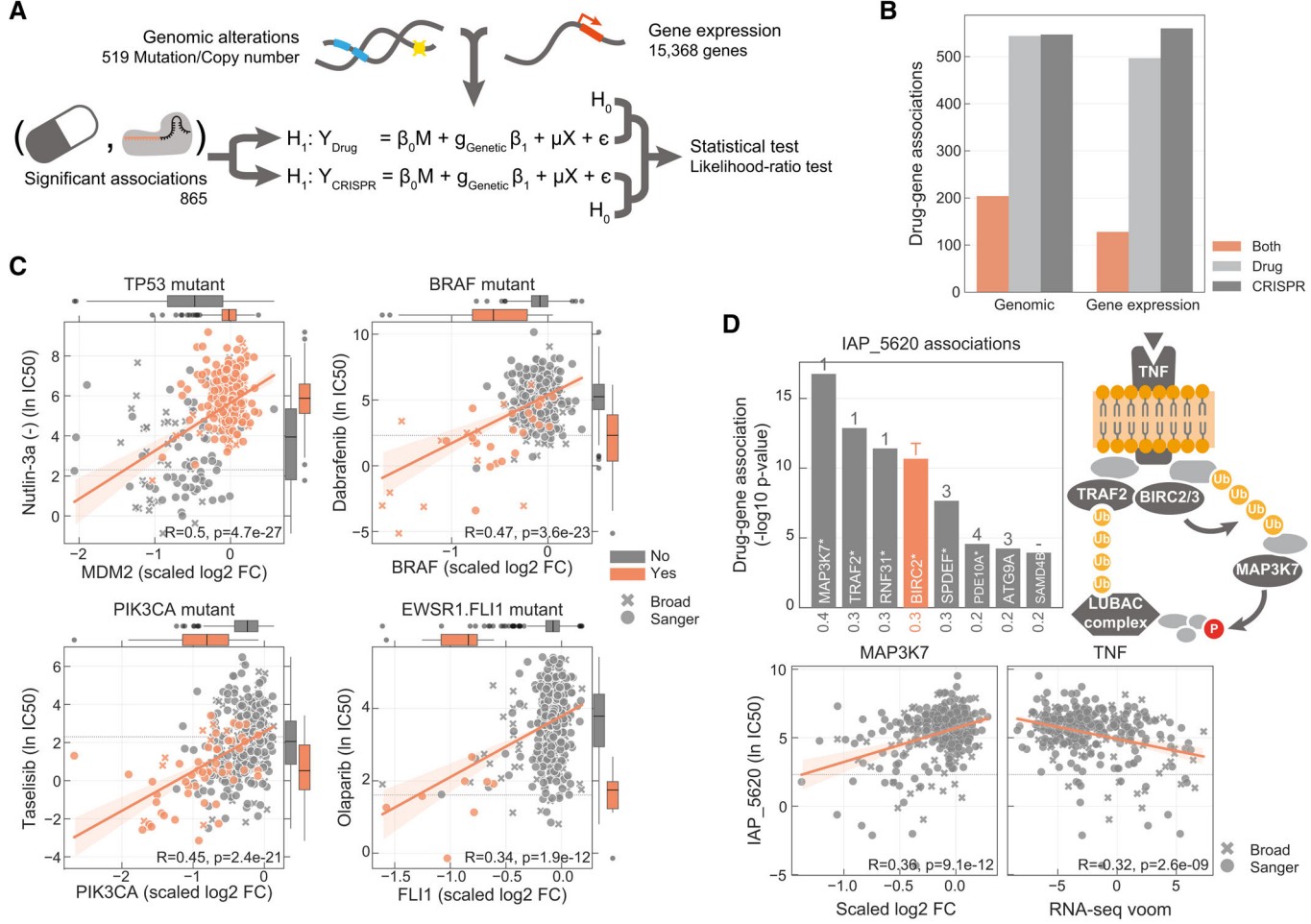

**Figure 5. Robust pharmacological associations.**

A  Diagram representing how genomic and gene expression data sets are integrated to identify significant associations with drug–gene pairs that were previously found to be significantly correlated.

B  Number of drug–gene pairs with at least one significant association with drug response, gene fitness or both, considering either genomic or gene expression profiles.

C  Canonical examples of robust pharmacological associations. Box-and-whisker plots show 1.5× interquartile ranges and 5–95[th] percentiles, centres indicate medians.

D  Representative example of a BIRC2/BIRC3 inhibitor, IAP_5620, showing the significant associations (FDR-adjusted likelihood-ratio test *P*-value < 10%) with CRISPR gene fitness profiles and their location in a representation of the TNF pathway. Bar plot is defined similarly to Fig 3.

based on CRISPR dependency data, for multiple members of the cIAP pathway, including *BIRC2*, *MAP3K7* and *RNF31* (Beug *et al*, 2012) (Fig 5D).

# Discussion

Understanding drug mechanism-of-action and the biological pathways underpinning drug response is an important step in preclinical studies. Here, we demonstrate how the integration of drug sensitivity and CRISPR-Cas9 gene fitness data can be used to inform on multiple aspects of drug mechanism in cells, including drug specificity, isoform selectivity and potency. Our analysis recapitulated drug targets for approximately a quarter of the drugs tested and for approximately another quarter revealed associations enriched for proteins closely related with the drug target(s). Critically, the strength of these associations reflects specificity and polypharmacology of the

cancer drugs, with highly selective and potent drugs showing the strongest associations with their nominal target. Significant drug–gene associations define networks of protein interactions that are functionally related with drug targets and underpin drug response. This revealed a previously unappreciated interaction between *MARCH5* and MCL1 inhibitors, with potential utility to derive predictive models of MCL1 inhibitor response across multiple cancer types, and particularly in solid tumours such as breast carcinomas. Robust pharmacogenomic biomarkers leveraged both data sets to provide refined biomarkers that are correlated with both drug response and biological networks. Interestingly, the networks we have defined can provide alternative targets that are functionally related with the drug target and mediate similar effects on cell fitness, potentially providing strategies for combination therapies to limit therapy resistance.

Preclinical biomarker development is an important step in drug discovery and is associated with increased success rates during clinical development (Nelson *et al*, 2015). Traditionally, this has been

performed by building predictive models of drug response using mutation, copy number and gene expression (Iorio *et al*, 2016; Tsherniak *et al*, 2017). Here we extended this approach and propose what we term as robust pharmacogenomic association—a drug response and gene fitness pair that are significantly correlated and are also both significantly related to the same molecular biomarker. This approach gives greater confidence in molecular biomarkers identified, since they are recapitulated using data from two orthogonal assays and provides markers at the level of the network. In addition, by focusing only on drugs involved in significant gene–drug pairs, we enrich for drugs most likely to have greater specificity, and thereby better enabling biomarker discovery.

Nearly half of the drugs did not have a significant association with gene fitness effects and may warrant further investigation. Possible explanations for this include the following: (i) drug polypharmacology which is difficult to deconvolute using single-gene knockout data; (ii) intrinsic difference between protein inhibition and knockout; (iii) a dosage-dependent response leading to incomplete inhibition of the drug target; (iv) functional redundancy between protein isoforms resulting in less penetrant effects with gene knockout; and (v) technical limitations of CRISPR-Cas9 screens such as the introduction of DNA double-strand breaks and sgRNA on-target and off-target effects variability across cancer cell lines.

We expect that some of these issues can be addressed by expanding this analysis to integrate other types of functional genomic screens, such as CRISPR inhibition, which might mimic drug inhibition more closely and does not introduce double-strand breaks.

We used CRISPR loss-of-function data sets to comprehensively study drug mechanism-of-action in cancer cells. CRISPR screening data are now available for many cell lines, and the profiling of compounds across cell line panels is already regularly performed, and so this approach could become a routine step during drug development. In particular, it is likely to have utility during the hit-to-lead or lead optimisation stages of drug development to select compound series with optimal potency and selectivity. It could also be useful for novel and uncharacterised cell active compounds, particularly if integrated with orthogonal experimental (e.g. such as kinobead assays) or computational approaches (e.g. drug pocket binding) to find direct targets and interrogate mechanisms-of-action. The utility of this approach is likely to expand as the availability of CRISPR knockout screening data, and other data sets such as CRISPR activation and inhibition, increases across ever larger collections of highly annotated cancer cell models. In conclusion, this study illustrates a new approach for investigating cellular drug mechanism-of-action that can be applied to multiple critical aspects of drug development.

# Materials and Methods

## Reagents and Tools table

| Software | |
| --- | --- |
| Python v3.7.3 | https://www.python.org/downloads/release/python-370/ |
| Matplotlib v3.1.0 | https://matplotlib.org/ (Hunter, 2007) |
| Seaborn v0.10.0 | https://seaborn.pydata.org/ (Waskom *et al*, 2020) |
| Limix v3.0.3 | https://docs.limix.io/ (preprint: Lippert *et al*, 2014; Casale *et al*, 2017) |
| Numpy v1.18.1 | https://numpy.org/ (van der Walt *et al*, 2011) |
| Scipy v1.4.1 | https://www.scipy.org/ (Virtanen *et al*, 2020) |
| Pandas v1.0.1 | https://pandas.pydata.org/ (McKinney, 2010) |
| Scikit-learn v0.21.2 | https://scikit-learn.org/ (Pedregosa *et al*, 2011) |
| Pydot v1.4.1 | https://pypi.org/project/pydot/ |
| Python-igraph v0.7.1 | https://igraph.org/python/ (Csardi & Nepusz, 2006) |
| Crispy (Cy) v0.4.6 | https://pypi.org/project/cy/ (Gonçalves *et al*, 2019) |

## Methods and Protocols

### Cancer cell lines panel

The 484 cancer cell lines used in this manuscript have been compiled from publicly available repositories as well as private collections and maintained following the supplier guidelines. STR and SNP fingerprints were used to ensure cell lines selected were genetically unique and matched those in public repositories (http://cancer.sanger.ac.uk/cell_lines/download). Detailed cell line model information is available through Cell Model Passports database (https://cellmodelpassports.sanger.ac.uk/) (van der Meer *et al*, 2019). Cell lines growth rate is represented as the ratio between the mean of the untreated negative controls measured at

day 1 (time of drug treatment) and the mean of the DMSO treated negative controls at day 4 (72 h post-drug treatment).

### High-throughput drug sensitivity

Experimental details of both GDSC1 and GDSC2 screens can be found in the Genomics of Drug Sensitivity in Cancer (GDSC) project (www.cancerRxgene.org) (Yang *et al*, 2013). Cell viability and dose response curve fitting models were previously described in detail (Iorio *et al*, 2016; Vis *et al*, 2016). Maximum screened drug concentrations (μM) are provided in Dataset EV1. Each compound was measured on average across 393 cell lines rendering a nearly complete matrix with only 14.2% missing values. All considered compounds displayed an $IC_{50}$ lower than half of the maximum

screened concentration in at least three cell lines. This ensures the compounds display an informative profile in at least a subset of cell lines. Drug nominal oncology target(s) annotation was manually curated from literature (Dataset EV1).

### Genome-wide CRISPR-Cas9 dropout screens

The CRISPR-Cas9 screens for the 484 cancer cell lines considered in this study (Dataset EV4) were assembled from two distinct projects, 320 were generated as part of Sanger DepMap Project Score (Behan *et al*, 2019) and 164 from the Broad DepMap version 19Q3 (Meyers *et al*, 2017; DepMap, 2019). Only cell lines that passed quality control filtering similarly to Behan *et al* (2019) and with matched drug response measurements were considered. Different CRISPR-Cas9 sgRNA libraries were used in each project (Koike-Yusa *et al*, 2014; Doench *et al*, 2016; Tzelepis *et al*, 2016). Consequently, library-specific effects were present (Dempster *et al*, 2019) (Fig EV2A) which hampers averaging of cell lines that were screened in both data sets. Thus, for the overlapping cell lines only data from Sanger DepMap Project Score was used. This also minimises potential cell line specific differences, for example due to genetic drift (Ben-David *et al*, 2018), and thereby increasing concordance with the drug response data set also generated at the Wellcome Sanger Institute. Fold changes (log$_2$) were estimated comparing samples with the respective control plasmid. Gene-independent deleterious effects induced by copy number amplifications in CRISPR-Cas9 screens (Aguirre *et al*, 2016; Munoz *et al*, 2016; Gonçalves *et al*, 2019) were corrected on a per sample basis using the unsupervised method CRISPRcleanR (Iorio *et al*, 2018). Replicates were mean averaged, and gene level fold changes were estimated by taking the mean of all the mapping sgRNAs. Gene level fold changes were quantile normalised per sample and then median scaled using previously defined lists of cancer cell lines essential and non-essential genes (Hart *et al*, 2015); thus, essential genes have a median log$_2$ fold change of $-1$ and non-essential genes a median log$_2$ fold change of 0. Only overlapping genes between the two libraries were considered, thus generating a full matrix of 16,643 genes across the 484 cell lines. A cell line was considered dependent on a gene if the knockout had a log$_2$ fold change of at least 50% of that expected of essential genes (scaled log$_2$ fold change $< -0.5$).

### PCA of drug sensitivity and gene fitness

Principal component analysis was performed using scikit-learn (Pedregosa *et al*, 2011) and the sklearn.decomposition.PCA class with default parameters and the number of components (n_components) set to 10. For the drug response data set, and only for the PCA analysis, missing values of each drug were imputed using the drug mean IC$_{50}$ response across the rest of the cell lines. Imputation was not required for the CRISPR-Cas9 data set since the matrix had no missing values.

### Drug response linear mixed model associations

Associations between drug response and gene fitness scores were performed using an efficient implementation of mixed-effect linear models available in the Limix Python module (preprint: Lippert *et al*, 2014; Casale *et al*, 2017). We considered the following covariates in the model: (i) binary variables indicating the institute of origin of the cell line CRISPR-Cas9 screen; (ii) principal component 1 of the drug response data set which is a correlative of cell lines

growth rate; and (iii) growing conditions (adherent, suspension or semi-adherent) represented as binary variables. Additionally, gene fitness similarity matrix of the samples is considered as random effects in the model to account for potential sample structure. Taken together, we fitted the following mixed linear regression model for each drug–gene pair:

$$d = \beta_0 M + \beta_1 e + \mu X + \varepsilon \qquad (1)$$

where $d$ represents a vector of the drug response IC$_{50}$ values across the cell lines; $M$ is the matrix of covariates; $\beta_0$ is the vector of effect sizes; e is the vector of gene CRISPR-Cas9 log$_2$ fold changes and $\beta_1$ the effect size; $X$ the similarity matrix based on the CRISPR-Cas9 gene fitness measurements; $\mu$ is the random effects; $\varepsilon$ is the general noise term. For each drug, cell lines with missing values were dropped from the fit.

We statistically assessed the significance of each association by performing likelihood-ratio tests between the alternative model $(\widehat{\theta_1})$ and the null model which excludes the CRISPR gene fitness scores vector $e$ and its parameter $\beta_1(\widehat{\theta_1})$. The parameter inference is performed using maximum likelihood estimation:

$$\widehat{\theta_1} = \text{argmax } p\ (d|M, X; \theta). \qquad (2)$$

And the P-value of the association is defined by:

$$\frac{p\left(d|M, X; \widehat{\theta_0}\right)}{p\left(d|M, X; \widehat{\theta_1}\right)}. \qquad (3)$$

We tested all the single-feature pairwise associations between the 480 compounds and the 16,643 genes, making a total of 7,988,640 tested associations. P-value adjustment for multiple testing was performed per drug using the Benjamini–Hochberg false discovery rate (FDR). Contrary to performing the adjustment across all tests, per drug correction has the following benefits: (i) associations assembled from the different screening platforms (GDSC1 and GDSC2) are kept separate hence not biasing for measurement type; and (ii) drugs with responses across larger subsets of cancer cell lines, for example Nutlin-3a response across TP53 wild-type cell lines, display stronger associations than most drugs; thus, correcting across all drugs would retain more associations from these drugs at a specific error rate, i.e. 10%, compared to the rest.

### Protein–protein interaction network

We assembled from the STRING database (Szklarczyk *et al*, 2017) a high confidence undirected protein–protein interaction network. We only consider interactions with a combined confidence score higher than 900. STRING identifiers were converted to HUGO gene symbols, and those not mapping or with multiple mappings were removed. Using *igraph* Python wrapper (Csardi & Nepusz, 2006), the network was simplified by removing unconnected nodes, self-loops and duplicated edges, leaving a total of 10,587 nodes and 205,251 interactions. A weighted version of the network was also assembled by correlating the gene fitness profiles of the connected nodes. Network nodes, and corresponding edges, that were not covered by the CRISPR-Cas9 screens were removed, making a total of 9,595 nodes and 172,584 weighted interactions.

### Robust pharmacogenomic associations

Robust pharmacological associations were estimated similarly to the previous associations, but in this case only drug–gene pairs that are significantly correlated were considered to test associations with the genomic features (binarised copy number and mutation status (Iorio *et al*, 2016)) and gene expression profiles (RNA-seq voom (Law *et al*, 2014) transformed RPKMs (Garcia-Alonso *et al*, 2018)). A robust pharmacogenomic association is defined as follows: (i) a drug–gene pair whose drug sensitivity and gene fitness is significantly correlated, and (ii) genomic alteration or gene expression profile is significantly correlated with both drug response and gene fitness. Log-ratio test *P*-values are independently estimated for drug response and gene fitness measurements and corrected per drug–gene. Drug–gene pairs where both are associated with a genomic or gene expression feature with an FDR lower than 10% are called robust pharmacogenomic associations (Datasets EV7 and EV8).

### Predictive models of drug response of MCL1 inhibitors

L2-regularised linear regression models to predict MCL1 inhibitors drug response were trained using gene fitness, gene expression measurements or both of canonical regulators of *MCL1*, namely *MARCH5*, *MCL1*, *BCL2*, *BCL2L1*, *BCL2L11*, *PMAIP1*, *BAX*, *BAK1*, *BBC3*, *BID*, *BIK*, BAD. For the 9 MCL1 inhibitors considered in this study, predictive models of drug response measurements were trained using Ridge regressions with an internal cross-validation optimisation of the regularisation parameter, implemented in Sklearn with RidgeCV class (Pedregosa *et al*, 2011). Additionally, drug response measurements are split randomly 1,000 times, where 70% of the measurements are for training the model and 30% are left out as a test set. Model's performance is quantified using the $R^2$ metric on the test set, comparing the predicted vs the observed drug response measurements.

### Unsupervised drug target annotation using ChEMBL bioactivity profiles

Curated therapeutic targets were extracted from the ChEMBL database (version 25) (Mendez *et al*, 2019) for each drug, where available. Additional targets (including potential off-targets) were identified from the *in vitro* bioactivity data in ChEMBL where the "target type" description was one of the following: "single protein," "protein complex," "protein complex group," "protein–protein interaction" or "protein family." Only assays where the target organism corresponds to "Homo sapiens" were considered.

In order to define whether a drug is considered active in an *in vitro* assay, only those situations where a pChEMBL value was defined were selected for inclusion in the subsequent analysis. The pChEMBL value is aimed to harmonise all the comparable measures of half-maximal responses (molar IC50, XC50, EC50, AC50, Ki, Kd, potency and ED50) on a negative logarithmic scale, and it is calculated only when the standard relation in an assay is known to be "=". Activity thresholds were defined according to the target protein family, based on the Illuminating the Druggable Genome consortium (IDG https://druggablegenome.net/ProteinFam) as follows: Kinases: ≤ 30 nM; GPCRs: ≤ 100 nM; Nuclear Receptors: ≤ 100 nM; Ion Channels: ≤ 10 μM; Others: ≤ 1 μM.

## Data availability

Source code, analysis reports and Jupyter notebooks are publicly available in GitHub https://github.com/EmanuelGoncalves/dtrace. Drug response and CRISPR-Cas9 measurements are included in Datasets EV3 and EV4, respectively, and available at FigShare https://doi.org/10.6084/m9.figshare.10338413.v1.

**Expanded View** for this article is available online.

## Acknowledgements

We would like to thank all members of the Translational Cancer Genomics team who provided useful insight, including Matthew Coelho for providing careful reading of the manuscript. We are grateful for the diligent contributions of Jon Winter-Holt in terms of compound identification and data clearance, Pedro Beltrao for helpful discussions and Danilo Horta for support integrating the Limix package. Work in M.J.G laboratory was funded by Wellcome (206194) and AstraZeneca.

## Author contributions

Conceptualisation: EG and MJG; Formal analysis: EG; Data curation: EG, CP, DvdM, ABa, HL, JTL, BS, CC, FI, SF and MJG; Drug response acquisition and processing: DvdM, TM, ABa, LR, WY, EL, JH, CT, CH, IM, FT, JM, ABe and HL; Drug annotation: EG, AS-C, GP, FMB, PJ, EAC, ARL, CC and MJG; Writing original draft preparation: EG and MJG; Writing, reviewing and editing: all authors; Visualisation: EG; Supervision: ABa, AL, JTL, BS, CC, FI, SF and MJG; Funding acquisition: SF and MJG.

## Conflict of interest

This work was funded in part by AstraZeneca. M.J.G. receives funding from AstraZeneca. M.J.G., F.I. and A.R.L. receive funding from Open Targets, a public–private initiative involving academia and industry. F.I. performs consulting for the CRUK-AstraZeneca Functional Genomics Centre. J.T.L., B.S., C.C. and S.F. are current employees of AstraZeneca and hold stock in AstraZeneca.

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
