## [Review Process File · Molecular Systems Biology]

Drug mechanism-of-action discovery through the integration of pharmacological and CRISPR screens

Emanuel Goncalves, Aldo Segura-Cabrera, Clare Pacini, Gabriele Picco, Fiona Behan, Patricia Jaaks, Elizabeth Coker, Donny van der Meer, Andrew Barthorpe, Howard Lightfoot, Tatiana Mironenko, Alexandra Beck, Laura Richardson, Wanjuan Yang, Ermira Lleshi, James Hall, Charlotte Tolley, Caitlin Hall, Iman Mali, Frances Thomas, James Morris, Andrew Leach, James Lynch, Ben Sidders, Claire Crafter, Francesco Iorio, Stephen Fawell, and Mathew Garnett

DOI: 10.15252/msb.20199405

Corresponding author(s): Mathew Garnett (mg12@sanger.ac.uk)

Review Timeline:	Submission Date:	18th Dec 19
	Editorial Decision:	14th Feb 20
	Revision Received:	7th Apr 20
	Editorial Decision:	7th May 20
	Revision Received:	14th May 20
	Accepted:	21st May 20

Editor: Jingyi Hou

Transaction Report:

14th Feb 2020

Manuscript Number: MSB-19-9405

Title: Drug mechanism-of-action discovery through the integration of pharmacological and CRISPR screens

Author: Emanuel Goncalves

Aldo Segura-Cabrera

Clare Pacini

Gabriele Picco

Fiona Behan

Patricia Jaaks

Elizabeth Coker

Donny van der Meer

Andrew Barthorpe

Howard Lightfoot

Andrew Leach

James Lynch

Ben Sidders

Claire Crafter

Francesco Iorio

Stephen Fawell

Mathew Garnett

Dear Dr. Garnett,

Thank you for submitting your work to Molecular Systems Biology. We have now heard back from two of the three reviewers who agreed to evaluate your manuscript. Since the recommendations of these two reviewers are quite similar, I prefer to make a decision now rather than further delaying the process. If we receive comments from reviewer #1 we will forward them to you so that you can address any further issues raised.

As you will see below, the reviewers acknowledge that the presented method and findings seem interesting. They raise however a series of concerns, which we would ask you to address in a major revision. The recommendations of the reviewers are rather clear and therefore there is no need to repeat the points listed below. Please feel free to contact me in case you would like to discuss in further detail any of the issues raised by the reviewers.

On a more editorial level, we would ask you to address the following issues:

- Please provide a .docx formatted version of the manuscript text (including legends for main figures, EV figures and tables). Please make sure that the changes are highlighted to be clearly visible.
- Please provide individual production quality figure files as .eps, .tif, .jpg (one file per figure).
- Please provide a .docx formatted letter INCLUDING the reviewers' reports and your detailed point-

by-point responses to their comments. As part of the EMBO Press transparent editorial process, the point-by-point response is part of the Review Process File (RPF), which will be published alongside your paper.

-Please note that all corresponding authors are required to supply an ORCID ID for their name upon submission of a revised manuscript.

- We have replaced Supplementary Information by the Expanded View (EV format). In this case, all additional Figures can be provided as EV Figures. Please provide the images as individual files and include the EV Figure legends in the main text together with the main Figure legends. For detailed instructions regarding Expanded View please refer to our Author Guidelines:
<https://www.embopress.org/page/journal/17444292/authorguide#expandedview>.

- The tables currently provided as Supplementary Tables 1-8 should be provided (and called out in the text) as Datasets EV1-EV8. Please provide each of them as an .xls files and include a brief description of the Dataset in a separate tab.

- Before submitting your revision, primary datasets (and computer code, where appropriate) produced in this study need to be deposited in an appropriate public database (see <https://www.embopress.org/page/journal/17444292/authorguide#dataavailability>). - Dataset #1
- Dataset #2>

The accession numbers and database should be listed in a formal "Data Availability " section (placed after Materials & Method) that follows the model below (see also <https://www.embopress.org/page/journal/17444292/authorguide#dataavailability>). Please note that the Data Availability Section is restricted to new primary data that are part of this study.

Data availability

- We would encourage you to include the source data for figure panels that show essential quantitative information. Additional information on source data and instruction on how to label the files are available at < <https://www.embopress.org/page/journal/17444292/authorguide#sourcedata> >.

- All Materials and Methods need to be described in the main text. We would encourage you to use 'Structured Methods', our new Materials and Methods format. According to this format, the Material and Methods section should include a Reagents and Tools Table (listing key reagents, experimental models, software and relevant equipment and including their sources and relevant identifiers) followed by a Methods and Protocols section in which we encourage the authors to

describe their methods using a step-by-step protocol format with bullet points, to facilitate the adoption of the methodologies across labs. More information on how to adhere to this format as well as downloadable templates (.doc or .xls) for the Reagents and Tools Table can be found in our author guidelines: <

<https://www.embopress.org/page/journal/17444292/authorguide#researcharticleguide>>. An example of a Method paper with Structured Methods can be found here: .

- Please provide a "standfirst text" summarizing the study in one or two sentences (approximately 250 characters, including space), three to four "bullet points" highlighting the main findings. I noticed that you have already provided a "synopsis image" in the pdf format. Please provide it in a jpeg format (550px width and max 400px height).

- When you resubmit your manuscript, please download our CHECKLIST (http://embopress.org/sites/default/files/Resources/EP_Author_Checklist.xls) and include the completed form in your submission. *Please note* that the Author Checklist will be published alongside the paper as part of the transparent process <http://msb.embopress.org/authorguide#transparentprocess>.

If you feel you can satisfactorily deal with these points and those listed by the referees, you may wish to submit a revised version of your manuscript. Please attach a covering letter giving details of the way in which you have handled each of the points raised by the referees. A revised manuscript will be once again subject to review and you probably understand that we can give you no guarantee at this stage that the eventual outcome will be favorable.

Yours sincerely,

Jingyi Hou
Editor
Molecular Systems Biology

If you do choose to resubmit, please click on the link below to submit the revision online *within 90 days*.

Link Not Available

IMPORTANT: When you send your revision, we will require the following items:

1. the manuscript text in LaTeX, RTF or MS Word format
2. a letter with a detailed description of the changes made in response to the referees. Please specify clearly the exact places in the text (pages and paragraphs) where each change has been made in response to each specific comment given
3. three to four 'bullet points' highlighting the main findings of your study
4. a short 'blurb' text summarizing in two sentences the study (max. 250 characters)
5. a 'thumbnail image' (550px width and max 400px height, Illustrator, PowerPoint or jpeg format), which can be used as 'visual title' for the synopsis section of your paper.
6. Please include an author contributions statement after the Acknowledgements section (see <https://www.embopress.org/page/journal/17444292/authorguide>)

7. Please complete the CHECKLIST available at (<http://bit.ly/EMBOPressAuthorChecklist>). Please note that the Author Checklist will be published alongside the paper as part of the transparent process

(<https://www.embopress.org/page/journal/17444292/authorguide#transparentprocess>).

8. Please note that corresponding authors are required to supply an ORCID ID for their name upon submission of a revised manuscript (EMBO Press signed a joint statement to encourage ORCID adoption). (<https://www.embopress.org/page/journal/17444292/authorguide#editorialprocess>)

Currently, our records indicate that there is no ORCID associated with your account.

Please click the link below to provide an ORCID:

Link Not Available

The system will prompt you to fill in your funding and payment information. This will allow Wiley to send you a quote for the article processing charge (APC) in case of acceptance. This quote takes into account any reduction or fee waivers that you may be eligible for. Authors do not need to pay any fees before their manuscript is accepted and transferred to the publisher.

REFeree REPORTS

Reviewer #2:

Review: Drug mechanism-of-action discovery through the integration of pharmacological and CRISPR screens

This manuscript by Goncalves et al. describes the integration of nearly 200k drug sensitivity measurements for 397 unique cancer drugs and genome-wide CRISPR loss-of-function screens in various 484 cell lines to systematically investigate drug mechanism-of-action (MOA) in cells. The proposed analytical methods in this study add the value to the concept of training computation models for predicting drug MOA and drug responses from existing data, with the novelty of using the CRISPR data (i.e. genetic fitness) to train the models.

This approach leverages pharmacological and CRISPR screening data, I wonder whether authors looked into if and how the drug-gene associations from this study relate to the drug-gene interactions detected from chemogenetic screens (CRISPR screens with the added treatment arm, i.e. combined chemical and genetic perturbations)?

Pg. 3. Re: 'Parallel integration of gene loss-of-function screens with drug response can be used to investigate drug mechanism-of-action ...'

There are other studies which integrated gene loss-of-function screens with drug response to investigate drug MOA besides the ones authors referenced here. Authors might consider referencing those studies as well to stay on track with the most recent relevant data content (e.g. Hustedt et al. 2019, Zimmermann et al. 2018, Wang et al. 2018, etc.)

Pg. 3. Re: 'We show that CRISPR-Cas9 datasets recapitulate drug targets, can provide insights into drug potency and selectivity, and define cellular networks underpinning drug sensitivity.'

Yes, authors show this in their manuscript, however, this is not a novel finding, similar statements have been previously reported.

Pg. 4. Re: '... was significantly correlated ...' - authors could specify that it is a negative correlation.

Pg. 6. Re: 'For 76 drugs no significant association with their target was identified ... Thus, 47.5% of the annotated compounds (n=170) has an association with either the target or a functionally-related protein.'

It is not clear to me about which 76 drugs authors are talking here; 76 drugs from 26% of the 358 drugs with target annotation and for which they identified significant drug-gene pairs with their putative targets (first bar in the Figure 1c), or 76 drugs from the remaining 74% of drugs which don't have a significant association with their target gene's knockout?

In addition to that, 47.5% of the annotated compounds (n=170), doesn't match the percentage portrayed in the panel Figure 1c.

Pg. 7. Re: '... pathway members that have strongly correlated fitness profiles, which are likely functionally related (Pan et al, 2018).'

Authors cited only one study for this statement. This has been characterized and studied by several groups, so authors might consider citing other related work (e.g. Wang et al. 2017, Boyle et al. 2018, Rauscher et al. 2018, Kim et al. 2019)

Pg. 7. Re: 'For EGFR inhibitors these included tyrosine receptor kinases NTRK3 and MET, and the protein phosphatase PTPN11 (Wang et al, 2017; Pan et al, 2018) (Figure 2d).'

Mentioned kinases are not shown in the referenced panel.

Pg. 7. and Pg. 8. Re: Supplementary Figure 3.

Labeling of panels is incorrect, as well as figure legends - not following the main text.

Pg. 9. Re: 'Similarly, we observed that selective EGRF inhibitors cetuximab, erlotinib and gefitinib (Figure 3) were associated with EGFR but not ERBB2, whereas ...'

Has the selectivity of these inhibitors been reported previously? How impactful are these observations on existing therapies for cancers stemming from alterations in EGRF and ERBB2?

Reviewer #3:

Refinement of the mechanism of action of anti-cancer therapeutics emerging from phenotypic or target-based screens is critical both for guiding a mechanistic understanding of drug efficacy and toxicity. Previous attempts to solve this challenge have largely relied on low-throughput biophysical-based measurements or the use of high-dimensional readouts to match gene and drug perturbations.

Here, Goncalves et al, attempt to tackle this challenge systematically in a new way by leveraging the recently published genome-wide CRISPR viability screening datasets that they and others have produced. Their premise in this proof-of-concept manuscript using established cancer drugs with largely known mechanisms of action that the correlation in viability between a genetic knockout and an established drug across nearly 500 cell lines should rediscover the MOA and/or shed new light on it.

Through an extensive series of supervised linear regression analyses they demonstrate the merits of this approach. They find that in 26% of cases, the killing pattern of drugs is directly phenocopied by the CRISPR killing pattern of the known drug target. They explore protein-protein interactions and find new relationships, and also look for "robust biomarkers" that independently explain both the CRISPR killing and drug killing. They discover an exciting relationship between the MARCH5 E3 ligase and MCL1 inhibition, a finding, that with further study should be very interesting for exploiting the MCL1 addiction in many human cancers.

Overall, the paper and approach is solid and interesting. The analysis approach is largely straightforward and while not completely novel, is applied here to a new dataset in a new way. The MARCH5 finding could ultimately be quite important. I believe that many readers will appreciate this approach to the MOA challenge in cancer and this paper will be highly read and cited.

Some suggestions:

(1) The paper focuses on established drugs, with potent efficacy and (mostly) highly refined mechanisms of action. While this is useful proof-of-concept, it is unclear ultimately how the approach will work where the real MOA challenge is, which is for compounds in development. This might become a key part of the discussion. Are the same relationships observed when the drug killing effect is weaker (ie. IC10 or 20)? This might be instructive for this future application.

(2) It is possible that straightforward linear mixed regression model analyses used here may miss signal that lies in the tail of the distribution (where the killing is), and overweight the bulk of the distribution. It may be helpful to compare and contrast several analytical approaches.

(3) From my read, it seems the paper is largely the product of supervised analysis (drug target or PPIs of target). Were unsupervised results explored, recognizing the need to correct for multiple hypotheses? Perhaps there is more room for discovery here?

(4) I think the manuscript could be strengthened a bit if the overly simplistic concept of a singular drug target of each drug was softened. This is alluded to in some places, but not others. When a CRISPR and drug profile correlate, this may be due to picking up signal of an additional "off-annotated target" effect, or a "pathway" effect of the intended target. How do we know which correlations are due to which effects?

Minor points:

(a) I think the Figures, especially Figure 3, could benefit from some additional design focus.

(b) For the 26% of drugs that match CRISPR, do we see enrichment for particular categories of drugs, or for those with broader spectra of killing, or strength of killing, etc? Have we learned any general lessons from these or is the result purely stochastic?

(c) I would be careful about this statement "hence, CRISPR measurements are more powered than

gene expression to identify drug functional interaction networks". The strength of this statement may overinterpret the given analysis.

Response to reviewers

We thank the reviewers for their constructive feedback on our manuscript and believe we have been able to address their comments. Below follows our point-by-point reply to the reviewers' comments (provided in italics), and our responses as well as significant changes to the manuscript text are highlighted in blue font.

""""

Reviewer #2:

This manuscript by Goncalves et al. describes the integration of nearly 200k drug sensitivity measurements for 397 unique cancer drugs and genome-wide CRISPR loss-of-function screens in various 484 cell lines to systematically investigate drug mechanism-of-action (MOA) in cells. The proposed analytical methods in this study add the value to the concept of training computation models for predicting drug MOA and drug responses from existing data, with the novelty of using the CRISPR data (i.e. genetic fitness) to train the models.

This approach leverages pharmacological and CRISPR screening data, I wonder whether authors looked into if and how the drug-gene associations from this study relate to the drug-gene interactions detected from chemogenetic screens (CRISPR screens with the added treatment arm, i.e. combined chemical and genetic perturbations)?

""""

We believe that CRISPR screens followed by drug treatment (with control arm) would compare more readily with multiplexed CRISPR screens where multiple genes are perturbed simultaneously. Indeed, this would be interesting to investigate. In this analysis, we look for drug and genetic perturbation effects that are introduced independently and correlate across the screened cell lines. To the best of our knowledge, chemogenetic screens across a large set of cell lines have not been performed (neither have multiplexed CRISPR screens) and so data to perform the proposed analyses are currently unavailable.

""

Pg. 3. Re: 'Parallel integration of gene loss-of-function screens with drug response can be used to investigate drug mechanism-of-action ...'

There are other studies which integrated gene loss-of-function screens with drug response to investigate drug MOA besides the ones authors referenced here. Authors might consider referencing those studies as well to stay on track with the most recent relevant data content (e.g. Hustedt et al. 2019, Zimmermann et al. 2018, Wang et al. 2018, etc.)

""

We apologize for omitting important references to the literature. We agree with the reviewer and have added the suggested references.

""

Pg. 3. Re: 'We show that CRISPR-Cas9 datasets recapitulate drug targets, can provide insights into drug potency and selectivity, and define cellular networks underpinning drug sensitivity.'

Yes, authors show this in their manuscript, however, this is not a novel finding, similar statements have been previously reported.

""

We agree with the reviewer's comment. We want to emphasise that from our comprehensive analysis integrating CRISPR-Cas9 screens revealed many functional aspects of cancer drugs that were not explored before. We rephrased this sentence accordingly. We have also now referenced previous studies using loss-of-function screens who have utilised a similar approach.

""

Pg. 4. Re: '... was significantly correlated ...' - authors could specify that it is a negative correlation.

""

Thank you for the suggestion and we have added this information.

""""

Pg. 6. Re: 'For 76 drugs no significant association with their target was identified ... Thus, 47.5% of the annotated compounds (n=170) has an association with either the target or a functionally-related protein.'

It is not clear to me about which 76 drugs authors are talking here; 76 drugs from 26% of the 358 drugs with target annotation and for which they identified significant drug-gene pairs with their putative targets (first bar in the Figure 1c), or 76 drugs from the remaining 74% of drugs which don't have a significant association with their target gene's knockout?

In addition to that, 47.5% of the annotated compounds (n=170), doesn't match the percentage portrayed in the panel Figure 1c.

""""

We apologize for the confusion; the text did not clearly define the drugs we were referring to. The reviewer is correct to say that the 76 drugs came from those 74% (n=264) of drugs which do not have a significant association with their target's knockout. To clarify, for 358 drugs we have information about their nominal targets which have also been knocked out in the CRISPR-Cas9 screens. From these, 26.3% have significant associations with the target and another 21.2% associations with genes closely related with the target (PPI distance 1, 2 and 3), making a total of 47.5% (n=170). We substantially rephrased the paragraph to make this clearer.

""""

Pg. 7. Re: '... pathway members that have strongly correlated fitness profiles, which are likely functionally related (Pan et al, 2018).'

Authors cited only one study for this statement. This has been characterized and studied by several groups, so authors might consider citing other related work (e.g. Wang et al. 2017, Boyle et al. 2018, Rauscher et al. 2018, Kim et al. 2019)

""""

We fully agree with the reviewer, these are important studies in the field that should have been referenced. We expanded the references accordingly.

""

Pg. 7. Re: 'For EGFR inhibitors these included tyrosine receptor kinases NTRK3 and MET, and the protein phosphatase PTPN11 (Wang et al, 2017; Pan et al, 2018) (Figure 2d).'

Mentioned kinases are not shown in the referenced panel.

""

We apologise for the inconsistency, only MET should have been mentioned, and due to previous cut-offs used to draw the sub-network of cetuximab, MET was not displayed. We have updated the figure and its legend, as well as the text.

""

Pg. 7. and Pg. 8. Re: Supplementary Figure 3.

Labeling of panels is incorrect, as well as figure legends - not following the main text.

""

We thank the reviewer for pointing this out. Supplementary Figure 3 references have been corrected and the figure panels ordered accordingly. The legend remains the same as it was in the correct order.

""

Pg. 9. Re: 'Similarly, we observed that selective EGRF inhibitors cetuximab, erlotinib and gefitinib (Figure 3) were associated with EGFR but nor ERBB2, whereas ...'

Has the selectivity of these inhibitors been reported previously? How impactful are these observations on existing therapies for cancers stemming from alterations in EGRF and ERBB2?

""

The selectivity of the molecules has been previously defined and erlotinib, gefitinib and cetuximab are all known to have selectivity for EGFR1 over other EGFR family members

(Source: <https://www.drugbank.ca/>). All three compounds are currently used in the clinic to inhibit EGFR across different cancers, specifically cetuximab is a monoclonal antibody approved for the clinical treatment of metastatic colorectal cancers (PMID:16117976), and erlotinib and gefitinib are small molecule inhibitors to treat non-small cell lung cancers (PMID:15329413, PMID:20573926). These findings are unlikely to have an impact on existing EGFR inhibitors in clinical use, but they would be an approach to confirm the selectivity of third and fourth generation EGFR inhibitors which may be developed, and they exemplify our ability to distinguish isoform selectivity as a proof-of-concept.

Reviewer #3:

Refinement of the mechanism of action of anti-cancer therapeutics emerging from phenotypic or target-based screens is critical both for guiding a mechanistic understanding of drug efficacy and toxicity. Previous attempts to solve this challenge have largely relied on low-throughput biophysical- based measurements or the use of high-dimensional readouts to match gene and drug perturbations.

Here, Goncalves et al, attempt to tackle this challenge systematically in a new way by leveraging the recently published genome-wide CRISPR viability screening datasets that they and others have produced. Their premise in this proof-of-concept manuscript using established cancer drugs with largely known mechanisms of action that the correlation in viability between a genetic knockout and an established drug across nearly 500 cell lines should rediscover the MOA and/or shed new light on it.

Through an extensive series of supervised linear regression analyses they demonstrate the merits of this approach. They find that in 26% of cases, the killing pattern of drugs is directly phenocopied by the CRISPR killing pattern of the known drug target. They explore protein-protein interactions and find new relationships, and also look for "robust biomarkers" that

independently explain both the CRISPR killing and drug killing. They discover an exciting relationship between the MARCH5 E3 ligase and MCL1 inhibition, a finding, that with further study should be very interesting for exploiting the MCL1 addiction in many human cancers.

Overall, the paper and approach is solid and interesting. The analysis approach is largely straightforward and while not completely novel, is applied here to a new dataset in a new way. The MARCH5 finding could ultimately be quite important. I believe that many readers will appreciate this approach to the MOA challenge in cancer and this paper will be highly read and cited.

””””

We thank the reviewer for their positive comments.

””””

(1) The paper focuses on established drugs, with potent efficacy and (mostly) highly refined mechanisms of action. While this is useful proof-of-concept, it is unclear ultimately how the approach will work where the real MOA challenge is, which is for compounds in development. This might become a key part of the discussion. Are the same relationships observed when the drug killing effect is weaker (ie. IC10 or 20)? This might be instructive for this future application.

””””

The reviewer raises an important application of this analysis. Firstly, we have not observed any substantial bias towards drugs with stronger drug responses among the significant drug - gene associations (Rebuttal Figure 1a). This could be in part explained by the fact that for this manuscript we only considered drugs that showed IC50s lower than 50% of the maximum screened concentration in at least 3 cancer cell lines. Additionally, we found that significant drug - gene associations are enriched for drugs with cyostatic/cytotoxic responses in a subset of cancer cell lines (Rebuttal Figure 1b). This suggests that to be able to identify drug - gene interactions, particularly drug - target, it is likely more important to have consistent responses in subsets of cell lines rather than the strength of the drug

response. Nonetheless, we cannot completely exclude that for drugs with weaker cytostatic/cytotoxic effects size will be smaller and thereby more difficult to capture their mode-of-action.

Rebuttal Figure 1. Drug response strength relation with drug - gene associations. *a*, for each drug, the difference between the minimum IC50 and the maximum concentration used for screening is reported in the x-axis, versus the strongest association found, i.e. lowest p-value (y-axis). *b*, for each drug the percentage of IC50 measurements across the 484 cancer cell lines that are lower than the maximum concentration used for screening is reported in the x-axis. The effect sizes of the drug-gene associations reported in the y-axis is represented in the size of the circles, i.e. stronger absolute associations correspond to larger circles. Drug-gene associations distance to nominal targets of the drugs in the protein-protein interaction network.

We expanded the discussion section to address these points, as suggested. Briefly, we believe for compounds with unknown mode-of-action this type of analysis can provide evidence of potential direct targets if a single CRISPR KO correlates strongly with the compound response across the same set of cancer cell lines. The true drug target could be among the top associations and therefore we expect that our approach can be used to guide complementary experimental (e.g. kinobead) and computational (e.g. drug pocket binding) methods for further validation. In the absence of significant associations, this is less informative but it still can support that the compound (if showing cellular activity) is likely mediating its response through engaging multiple targets. We also believe our approach

could be useful for drugs in advanced development (e.g. hit or lead optimisation) to identify potential undesirable off-target activities, particularly for non-kinase off-target activities.

““““

(2) It is possible that straightforward linear mixed regression model analyses used here may miss signal that lies in the tail of the distribution (where the killing is), and overweight the bulk of the distribution. It may be helpful to compare and contrast several analytical approaches.

””””

The reviewer is right to point out that fundamental limitations of simple linear regressions and specific characteristics of drug response distributions might lead to miss some drug - gene associations. While we agree with this, some aspects of linear models also make them very well suited for this analysis:

- i. scalability, implementations of linear mixed models have been extensively optimised to handle hundreds to millions of tests, for example for eQTL analyses, this is important for this study as we performed a total of ~8 million tests;
- ii. availability of well-established and computationally efficient statistical tests, such as likelihood-ratio tests, that support comparisons with covariates and random effects and thereby are instrumental to statistically assess the added value of each gene CRISPR fitness profiles over potential confounding effects;
- iii. despite relying on the identification of simple linear associations, these approaches approximate reasonably well to drug - gene associations that deviate from that (Figure 5c and Rebuttal Figure 2);
- iv. we have previously tested non-linear functions (using python package `scipy curve_fit` function) and the associations found were largely overlapping with our systematic linear regression approach;
- v. outlier handling is a general problem across many analytical approaches, including linear models. Nonetheless, from several examples shown in Figure 1d, Figure 5c and

Rebuttal Figure 2 we believe these models are robust enough to identify associations in small subsets of samples even if they are mostly driven by values on the tail of the drug response;

- vi. prior to fitting the linear models, we standardise the drug response measurements by removing the mean and scaling to unit variance, this is a common procedure to many machine learning approaches to make the data ranges more comparable.

Taken together, despite the intrinsic limitations of linear regression models, we believe they provide a very flexible and scalable approach to identify relevant associations between drug response and CRISPR gene essentiality profiles.

Rebuttal Figure 2. Representative examples of drug - target associations of drugs with cell cytotoxic/cytostatic responses only present in a small subset of outlier cancer cell lines. MET inhibitor (left) and FGFR2 inhibitor (right) with response profiles in a small subset of the cell lines that show significant associations with their targets.

""""

(3) From my read, it seems the paper is largely the product of supervised analysis (drug target or PPIs of target). Were unsupervised results explored, recognizing the need to correct for multiple hypotheses? Perhaps there is more room for discovery here?

""""

We have taken an unsupervised approach to the identification of drug - gene associations. Thus, for the 480 cancer drugs (397 unique) we tested all possible associations with the 16,643 CRISPR KO genes, making a total of approximately 8 million tests. These associations were then interpreted using the nominal target annotations and the protein-protein interaction network. Supplementary Table 5 contains the 865 drug - gene significant associations identified in this study, and we also deposited in figshare (<https://10.6084/m9.figshare.10338413>) all ~8 million associations tested along with their effect sizes, statistical values and drug-target and PPI annotation. As the reviewer points out, multiple hypotheses testing becomes a challenge due to the large number of tests. Considering that drug measurements come from different technological approaches, drug - gene associations p-values were adjusted on a per drug basis, not overall. We observed that this helps finding the most relevant associations of each drug without the problem of enriching for drugs with an overall higher number of associations, such as Nutlin3-a and FGFR1 inhibitors. We have rephrased initial parts of the manuscript to make this more explicit.

““““

(4) I think the manuscript could be strengthened a bit if the overly simplistic concept of a singular drug target of each drug was softened. This is alluded to in some places, but not others. When a CRISPR and drug profile correlate, this may be due to picking up signal of an additional "off-annotated target" effect, or a "pathway" effect of the intended target. How do we know which correlations are due to which effects?

””””

We believe that multiplexed CRISPR screens (e.g. dual and triple knockouts) would be necessary to precisely identify targets of drugs with polypharmacology effects. This would likely inform on the portion of drugs (46.6%) for which we have not identified any significant association with single gene knockout. Unsupervised search of drug targets using ChEMBL bioactivity profiles showed that, despite low mean differences, drugs with significant drug-target association had lower number of putative targets (two-sided Welch's t-test p-value =

0.003) (new panel in Fig EV3d). We agree with the reviewer that it is important to understand when a drug - gene association is due to direct physical inhibition or indirect association of the drug response pathway. It is challenging to know this from our analysis alone and more evidence is generally required. Nonetheless, we believe our analysis can provide important insights to guide this interpretation. On the one hand, if the effects are related to “pathway” effects then these will be closely connected and functionally related to the known nominal targets of the drugs in the PPI network (PPI shortest path ≤ 3). We confirmed this is the case for the majority for the drugs we screened. On the other hand, if it is a potential off-target we expect no immediate link of the associated gene with any of the canonical targets of the drug. For example, ibrutinib, a BTK inhibitor, strongly correlates with EGFR and ERBB2 and is supported by kinobead measurements. The relative strength of the association can also provide important insights. For example, MCL1 inhibitors strongly correlate with MCL1 suggesting very selective associations, nonetheless MARCH5 is also related but more weakly. Thus, rather than a putative off-target MARCH5 is likely functionally related, even though String PPI does not have any relation between the two. This was confirmed independently by dual knockout screens, showing a synthetic-lethal interaction of BCL1L2 and MARCH5 (PMID:32029722). We thank the reviewer for this comment as it pointed out important aspects of the interpretation of our analysis that we did not consider in the discussion, and we expanded the discussion accordingly.

““““

Minor points:

(a) I think the Figures, especially Figure 3, could benefit from some additional design focus.

””””

We changed Figure 3 to group drug - gene associations by drug target classes, this way it corresponds better with the references in the text and makes it easier to compare the activity of drugs of the same target class.

““““

(b) For the 26% of drugs that match CRISPR, do we see enrichment for particular categories of drugs, or for those with broader spectra of killing, or strength of killing, etc? Have we learned any general lessons from these or is the result purely stochastic?

''''

Some drug target classes seem to be more predominantly represented in the group of 26% of drugs with significant correlation with their nominal targets (Rebuttal Figure 4). Nonetheless, due to the low number of drugs per target class we can not exclude these are biased by the specific set of drugs that we considered in this analysis. Drugs with a broad spectrum of killing across all cell lines are less likely to have significant associations with their targets (Rebuttal Figure 1b), and this seems to be less driven by cell killing strength (Rebuttal Figure 1a). In general, we observed that significant drug - target associations are more likely to be selective based on independent kinobead assay data (Figure 1e).

Rebuttal Figure 3. Significant drug - target associations grouped by drug target classes. Only drug target classes with at least 3 drugs were considered. Circle size is proportional to the number of drugs in that class with significant drug - target association.

““““

(c) I would be careful about this statement "hence, CRISPR measurements are more powered than gene expression to identify drug functional interaction networks". The strength of this statement may overinterpret the given analysis.

””””

We have rephrased this statement.

7th May 2020

Manuscript Number: MSB-19-9405R

Title: Drug mechanism-of-action discovery through the integration of pharmacological and CRISPR screens

Author: Emanuel Goncalves

Aldo Segura-Cabrera

Clare Pacini

Gabriele Picco

Fiona Behan

Patricia Jaaks

Elizabeth Coker

Donny van der Meer

Andrew Barthorpe

Howard Lightfoot

Tatiana Mironenko

Alexandra Beck

Laura Richardson

Wanjuan Yang

Ermira Lleshi

James Hall

Charlotte Tolley

Caitlin Hall

Iman Mali

Frances Thomas

James Morris

Andrew Leach

James Lynch

Ben Sidders

Claire Crafter

Francesco Iorio

Stephen Fawell

Mathew Garnett

Dear Dr Garnett,

Thank you for sending us your revised manuscript. We have now heard back from the two reviewers who were asked to evaluate your study. As you will see the reviewers are overall satisfied with the modifications made and think that the study is now suitable for publication.

Before we can formally accept your manuscript, we would ask you to address a few remaining issues listed below:

1. Please address/reply to reviewer #3's comment on the duplicated information in Fig3A and Fig4A.

On a more editorial level, please do the following:

- Author contributions: please make a differentiation between Andrew Barthorpe and Alexandra

Beck.

- Please remove the Dataset legends and figures from the main manuscript, only figure legends should stay in.

- I notice that you have already provided a synopsis image. Can you resize it into a smaller figure (550px width and ~400px height) and make sure that the text still has a decent resolution after resizing?

Also, please remove the synopsis image (and text) from the main text.

- I have slightly modified the synopsis text. Please let me know if you are fine with it or if you would like to introduce further modifications.

Synopsis text:

This study integrates pharmacological and CRISPR screens in 484 cancer cell lines to systematically investigate anticancer drug mechanism of action, yielding insights into the genetic contexts and cellular networks underpinning drug response.

- CRISPR screens reveal important aspects of drug mechanism-of-action, specifically in the context of cellular activity, isoform specificity, off-target and polypharmacological effects.
- By leveraging protein interaction networks that underlie drug-responses, novel drug-target interactions involving anti-apoptotic MCL1 inhibitors are identified.
- Improved pharmacogenomic biomarker discovery using two independent and orthogonal cell viability screens.

Please resubmit your revised manuscript online, with a covering letter listing amendments and responses to each point raised by the referees. Please resubmit the paper ****within one month**** and ideally as soon as possible. If we do not receive the revised manuscript within this time period, the file might be closed and any subsequent resubmission would be treated as a new manuscript. Please use the Manuscript Number (above) in all correspondence.

When you resubmit your manuscript, please download our CHECKLIST (<http://bit.ly/EMBOPressAuthorChecklist>) and include the completed form in your submission.

Please note that the Author Checklist will be published alongside the paper as part of the transparent process

(<https://www.embopress.org/page/journal/17444292/authorguide#transparentprocess>)

Click on the link below to submit your revised paper.

Link Not Available

Yours sincerely,

Jingyi Hou
Editor

If you do choose to resubmit, please click on the link below to submit the revision online before 6th Jun 2020.

Link Not Available

IMPORTANT: When you send your revision, we will require the following items:

1. the manuscript text in LaTeX, RTF or MS Word format
2. a letter with a detailed description of the changes made in response to the referees. Please specify clearly the exact places in the text (pages and paragraphs) where each change has been made in response to each specific comment given
3. three to four 'bullet points' highlighting the main findings of your study
4. a short 'blurb' text summarizing in two sentences the study (max. 250 characters)
5. a 'thumbnail image' (550px width and max 400px height, Illustrator, PowerPoint or jpeg format), which can be used as 'visual title' for the synopsis section of your paper.
6. Please include an author contributions statement after the Acknowledgements section (see <https://www.embopress.org/page/journal/17444292/authorguide#manuscriptpreparation>)
7. Please complete the CHECKLIST available at (<http://bit.ly/EMBOPressAuthorChecklist>). Please note that the Author Checklist will be published alongside the paper as part of the transparent process (<https://www.embopress.org/page/journal/17444292/authorguide#transparentprocess>).
8. Please note that corresponding authors are required to supply an ORCID ID for their name upon submission of a revised manuscript (EMBO Press signed a joint statement to encourage ORCID adoption) (<https://www.embopress.org/page/journal/17444292/authorguide#editorialprocess>).

Currently, our records indicate that the ORCID for your account is 0000-0002-2618-4237.

Link Not Available

The system will prompt you to fill in your funding and payment information. This will allow Wiley to send you a quote for the article processing charge (APC) in case of acceptance. This quote takes into account any reduction or fee waivers that you may be eligible for. Authors do not need to pay any fees before their manuscript is accepted and transferred to the publisher.

REFEREE REPORTS

Reviewer #2:

Authors have addressed all concerns. This is a cleanly written paper with strong support for its conclusions.

Reviewer #3:

The authors have done an excellent job revising the manuscript and addressing the reviewers comments and suggestions.

The paper now seems suitable for publication with the one tiny exception that Fig 3A and 4A seem to be accidentally duplicated panels.

We have addressed the outstanding reviewer comment. Our response as well as significant changes to the manuscript text are highlighted in blue font.

“““

Reviewer #3:

The paper now seems suitable for publication with the one tiny exception that Fig 3A and 4A seem to be accidentally duplicated panels.

”””

We agree with the reviewer that some overlapping information is shown in both panels but respectfully argue that there are important differences in the panels and the duplication improves readability. Specifically, Fig 3A shows multiple MCL1 inhibitors as well as the BCL2 inhibitor venetoclax, while Fig 4A focuses solely on MCL1 inhibitors. In Fig 4A, we plotted all MCL1 inhibitors screened (10 compounds - AZD5991 has been screened twice) including those that did not have significant associations with CRISPR (MIM1 and UMI-77), and which are not plotted on Fig 3A. As presented, Fig 4A is more consistent with panels Fig 4D and 4E, which also provide information for all MCL1 inhibitors. For these reasons, we would like to keep both panels to retain clarity and consistency.

Corresponding Author Name: Dr. Mathew Garnett

Manuscript Number: MSB-19-9405